# Costs of providing HIV care and optimal allocation of HIV resources in Guyana

Chutima Suraratdecha[1], Robyn M. Stuart[2,3]*, Morris Edwards[4], Rhonda Moore[4], Nadia Liu[4], David P. Wilson[2,5,6,7], Rachel Albalak[8]

1 Division of Global HIV and TB, U.S. Centers for Disease Control and Prevention, Atlanta, Georgia, United States of America, 2 Department of Mathematical Sciences, University of Copenhagen, Copenhagen, Denmark, 3 Burnet Institute, Melbourne, Australia, 4 Ministry of Public Health, Georgetown, Guyana, 5 Monash University, Melbourne, Australia, 6 Kirby Institute, University of New South Wales, Sydney, Australia, 7 Department of Microbial Pathogenesis, University of Maryland, Baltimore, United States of America, 8 U.S. Centers for Disease Control and Prevention, Caribbean Region Office, Barbados, Santo Domingo, Dominican Republic

* robyn@math.ku.dk

## Abstract

### Introduction

Great strides in responding to the HIV epidemic have led to improved access to and uptake of HIV services in Guyana, a lower-middle-income country with a generalized HIV epidemic. Despite efforts to scale up HIV treatment and adopt the test and start strategy, little is known about costs of HIV services across the care cascade.

### Methods

We collected cost data from the national laboratory and nine selected treatment facilities in five of the country's ten Regions, and estimated the costs associated with HIV testing and services (HTS) and antiretroviral therapy (ART) from a provider perspective from January 1, 2016 to December 31, 2016. We then used the unit costs to construct four resource allocation scenarios. In the first two scenarios, we calculated how close Guyana would currently be to its 2020 targets if the allocation of funding across programs and regions over 2017–2020 had (a) remained unchanged from latest-reported levels, or (b) been optimally distributed to minimize incidence and deaths. In the next two, we estimated the resources that would have been required to meet the 2020 targets if those resources had been distributed (a) according to latest-reported patterns, or (b) optimally to minimize incidence and deaths.

### Results

The mean cost per test was US$15 and the mean cost per person tested positive was US$796. The mean annual cost per of maintaining established adult and pediatric patients on ART were US$428 and US$410, respectively. The mean annual cost of maintaining virally suppressed patients was US$648. Cost variation across sites may suggest opportunities for improvements in efficiency, or may reflect variation in facility type and patient volume. There may also be scope for improvements in allocative efficiency; we estimated a 28% reduction

**Data Availability Statement:** All relevant data are available from GitHub (github.com/optimamodel/hiv_guyana).

**Funding:** Funding was provided by the Centers for Disease Control and Prevention, the President's Emergency Plan for AIDS Relief (PEPFAR) through the Centers for Disease Control and Prevention under the terms of contract numbers 200-2017-F-92918, GH15-1539 and GH001351, and the Australian National Health and Medical Research Council.

**Competing interests:** The authors declare no competing interests.

in the total resources required to meet Guyana's 2020 targets if funds had been optimally distributed to minimize infections and deaths.

## Conclusions

We provide the first estimates of costs along the HIV cascade in the Caribbean and assessed efficiencies using novel context-specific data on the costs associated with diagnostic, treatment, and viral suppression. The findings call for better targeting of services, and efficient service delivery models and resource allocation, while scaling up HIV services to maximize investment impact.

## Introduction

Guyana, a lower-middle-income country with per capita income of US$4,979 in 2018 [1], is part of the Caribbean region situated on the northern mainland of South America. Guyana's HIV epidemic is categorized as generalized, with higher prevalence among key populations (6.1% among female sex workers (FSW), 4.9% among men having sex with men (MSM)) [2]. Great strides in responding to the HIV epidemic have led to improved epidemiological outcomes: Guyana's 2015 AIDS Response Progress Report (the latest such report with detailed epidemic estimates) showed a steady decrease in the number of newly diagnosed HIV (1,176 in 2009 to 751 in 2014) and AIDS cases (1,219 in 2009 to 856 in 2014), along with a decline in both the proportion of deaths attributable to AIDS (9.5% in 2002 to 4.8% in 2012) and in adult HIV prevalence (2.4% in 2004 to 1.4% in 2013; 2018 estimates from UNAIDS also indicate prevalence of 1.4% [2]). Despite treatment scale-up, gaps remain, with only ~35% of people living with HIV (PLHIV) virally suppressed in 2018 [3]. Geographically, Region 4 (home to the capital city of Georgetown and to ~42% of the country's population) has historically had the majority of reported HIV cases [3].

The Guyana National AIDS Programme Secretariat (NAPS) has led the response to the HIV epidemic in collaboration/partnership with external stakeholders, primarily the Global Fund to Fight AIDS, Tuberculosis, and Malaria (GFATM) and the United States President's Emergency Plan for AIDS Relief (PEPFAR). Cost considerations for officially adopting the World Health Organization (WHO) "Test and Start" ART guidelines (treat-all), which recommend immediate initation on ART after diagnosis, will include costs associated with HIV testing services (HTS), ART, and viral load (VL) monitoring. With limited information on the costs of providing testing and treatment services in Guyana, making accurate estimates of resource needs is challenging. In addition, limited resources call for optimizing resource allocation to meet the program goals.

This study aims to fill the gaps in the economic data by assessing the economic costs and cost drivers of providing HIV services along the cascade of care in Guyana. The study also investigates the degree to which allocative efficiencies of HIV response could be exploited in order to reduce the costs associated with achieving Guyana's epidemiological targets using the estimates of the per-person cost along the HIV cascade.

## Methods

The study was approved by the Institutional Review Board Guyana Ministry of Public Health (MOPH) and the U.S. Centers for Disease Control and Prevention (CDC). All procedures followed were in compliance with the Helsinki Declaration, the Code of Federal Regulations,

Title 45, Part 46 (45 CFR §46), and local ethical and legal requirements. Consent was not required for this study.

To assess the economic costs associated with HTS and ART delivered by facility-based staff from a programmatic (provider) perspective, we collected data retrospectively at nine of twenty-two treatment facilities (public and private hospitals, and health clinics) in 5 of 10 Regions (2, 3, 4, 6 and 10) where most HIV cases were reported, and the national public health laboratory (NPHL) that provides laboratory testing to the whole country from January 1 to December 31, 2016. The nine facilities were purposively selected aiming to capture the range of costs across HIV diagnosis and treatment service delivery models associated with testing, linkage, treatment, and clinical monitoring of treatment. HIV services provided by these facilities accounted for 87.2% of national ART patients.

According to the Guyana HIV management guidelines, Guyana tests adult patients using the serial HIV rapid-testing algorithm with Determine™, Unigold™, and Statpak™ kits [4]. HIV Deoxyribonucleic Acid polymerase chain reaction (DNA PCR) testing was introduced in August 2008 for children under 18 months of age [5]. To link HIV positives to care, counselors discuss CD4 testing and the impact of early treatment with positive clients. Comprehensive treatment includes supportive care, laboratory monitoring, regular clinical assessments, and a combination of three or more antiretroviral drugs (ARV). According to the Guyana's HIV treatment guidelines [4], each patient should receive at least two CD4 tests and two VL tests annually. Retention and adherence activities may include counseling prior to initiating ART, additional counseling at every visit, monitoring and follow-up with patients, and an adherence support system (community and/or peer-support). Patients who unjustifiably miss more than two successive appointments are flagged for more rigorous adherence support (e.g., social support mechanisms, intensive adherence counseling). At the time that this study was conducted, patients not meeting the ART initiation eligibility criteria of CD4 count < 500 were labeled "pre-ART" and provided the same clinical and laboratory services (sometimes at a different frequency) without ARV, and pre-ART patients who subsequently met the eligibility criteria would be initiated on ART. Since the study time period, plans have been put into place (and to some extent, enacted) to implement the "treat-all" strategy recommended by the WHO, but the official guidelines have not yet been adjusted to reflect this [4].

The modified micro-costing and bottom-up approaches were used to collect cost data by input type (recurrent costs: personnel, ARV and other drugs, test kits, lab and other supplies, building use, utilities and contract services; capital costs: equipment, training and infrastructure). We conducted fieldwork from February through March 2017 collecting data on program costs and beneficiary volume at each study site through the extraction from existing data systems and interviews with facility personnel. Capital costs (training, buffer stock, equipment, and building construction/renovation) were annuitized over the estimated useful life of each cost item, with a discount rate of 3%, consistent with accepted methods [6]. All costs were collected in local currency (Guyanese dollar) and converted to U.S. dollars (USD) at the average market exchange rate of 1USD = 206.5 Guyanese dollars. Unit costs are estimated by dividing total costs (HTS, ART) by the relevant outcome measure (number of tests, number of tested positives, number of pre-ART or ART patients, number of virally suppressed patients). Analyses were conducted using the Stata SE14 [7] data analysis and statistical software package.

To assess the epidemiological impact and allocative efficiency of the HIV response, we used the Optima HIV tool [8], a compartmental model of HIV transmission and disease progression linked to a programmatic response module for estimating the epidemiological and economic impacts of interventions [9,10]. We created a model of Guyana's population disaggregated into 10 groups (males and females aged 15–24, 25–49 and 50+ years, FSW, clients of FSW, MSM, and children aged 0–14 years). We populated the model with available

data on population sizes, HIV testing rates, care and treatment coverage and sexual behavior (Table 1), and non-context specific parameters documented in the Optima HIV user guide [11]. We then fit the model to historical HIV prevalence data (Table 1) to produce baseline estimates of epidemic indicators (including PLHIV, infections, and deaths) up to 2016.

Estimates and projections of HIV epidemic indicators for 2017 and beyond take into account the reported HIV spending in 2015 (the latest year for which spending data were available) and its distribution across programmatic categories (Table 2), and unit costs for HTS and ART generated in this study. HIV investments in Guyana for 2015 totaled US$10.5m. This was lower than in previous years (US$17.0m in 2011, US$27.0m in 2012) [16]. Around 64% of the response was funded by PEPFAR, 29% domestically and 7% by the GFATM.

We considered Guyana's prospects for meeting its National Strategic Plan (NSP) 2020 targets [12]. Whilst progress towards these targets is not yet known, we estimated the likely progress under four alternative scenarios. The first two scenarios assume that the 2015 budget envelope was annually available over 2017–2020 and was allocated (a) according to latest-reported investment patterns, and (b) across different programs (Table 2) and geographical regions to minimize cumulative infections and deaths, using the geographical optimization algorithm described in [21]. In brief, the algorithm has two main steps: (1) for each region, take a range of different budget levels and for each budget level, calculate the allocation of funds across programs that would minimize cumulative infections and deaths; (2) find the allocation of the total national budget envelope among regions that minimizes overall infections and deaths. Finally, we create two additional scenarios in which we estimated how much additional funding would have been required to achieve Guyana's 2020 NSP targets if the funds were (a) allocated according to 2015 investment patterns, or (b) allocated in order to minimize HIV incidence and HIV-related deaths. These scenarios were chosen in consultation with Guyana's Ministry of Public Health; whilst none of these scenarios reflects a likely funding outcome, by comparing them we can obtain estimates on the possible scope for improvements in allocative efficiency.

## Results

Table 3 summarizes selected characteristics of the 9 study sites. The number of clients tested (tested positive) per site varied from 258 (3) to 3,476 (71) with an average testing yield of 2.6% (range: 0.9%-4.7%), and the number of patients on ART by site ranged from 30 to 1,686.

### Unit costs of HIV testing and services

HTS offered and costed at study sites include provider initiated testing and counseling, and outreach and linkage activities. The mean cost per test was US$14.7 and unit costs ranged from US$4.7 to US$31.4 (standard deviation (SD): 10.0) (Table 4). On average, personnel costs were the largest component (68%) of the costs of providing HTS, followed by utilities and supplies. The costs per positive client were associated with the testing yield. The mean costs per tested positive were US$796.1 and unit costs ranged from US$174.7 to US$3,548.0 (SD: US$1,053.3).

### Unit costs of treatment

Table 5 presents unit costs for each patient type (pre-ART, newly initiated ART adult, established ART adult, newly initiated ART pediatric and established ART pediatric). The average costs per patient year (PPY) for pre-ART and pediatric ART were lower than those for adult ART. Newly initiated ART adults (US$471.8) incurred higher average cost PPY than established patients (US$428.2), whereas the average cost PPY initiated pediatric (US$374.0) was

**Table 1. Model inputs.**

| Input | Value | Source |
|---|---|---|
| **Population size** | | |
| FSW | 3800 (2014); 5300 (2016) | [12,13] |
| Clients | 27045 (2014) | [12] |
| MSM | 2464 (2014); 3300 (2016) | [12,13] |
| Children 0–14 | 220974 (2015) | [14] |
| Males 15–24 | 84076 (2015) | [14] |
| Females 15–24 | 83348 (2015) | [14] |
| Males 25–49 | 116943 (2015) | [14] |
| Females 25–49 | 110385 (2015) | [14] |
| Males 50+ | 72710 (2015) | [14] |
| Females 50+ | 78649 (2015) | [14] |
| **HIV prevalence (%)** | | |
| FSW | 26.6 (2005); 16.6 (2009); 5.5 (2014) | [12] |
| Clients | 6.5 (2003); 3.8 (2009); 1.15 (2014) | [12,15] |
| MSM | 21.2 (2005); 19.4 (2009); 4.9 (2014) | |
| Children 0–14 | No data | [12] |
| Males 15–24 | 0.6% (2014) | |
| Females 15–24 | 0.9% (2014) | [16] |
| Males 25–49 | 1.9% (2014) | [16] |
| Females 25–49 | 2.1% (2014) | [16] |
| Males 50+ | No data | [16] |
| Females 50+ | No data | |
| **Other epidemiology** | | |
| Crude death rate | 8.137/1000 (2014) | [17] |
| TB prevalence | 131/100000 (2012) | [18] |
| **HIV testing rates (%)** | | |
| FSW | 84 (2011); 63.2 (2014) | [12,13] |
| Clients | 35.5 (2014) | |
| MSM | 72.3 (2011); 37.8 (2014) | [12] |
| Children 0–14 | No data | [12,13] |
| Males 15–24 | 13.5 (2009) | |
| Females 15–24 | 21.9 (2009) | |
| Males 25–49 | 24.3 (2009) | [19] |
| Females 25–49 | 30.3 (2009) | [19] |
| Males 50+ | No data | [19] |
| Females 50+ | No data | |
| **Care and treatment** | | |
| Number receiving ART | 4791 (2016); data for 2003–2014 from | |
| Women receiving PMTCT | [17] | [10] |
| Annual percentage lost to follow-up | 167 (2015) | [20] |
| Number of VL tests done | 7.5 (2014) | [20] |
| | 3482 (2014) | |
| **Sexual behavior** | | |
| Condom use: MSM, regular partners (%) | 79.7 (2009) | [20] |
| Condom use: MSM, casual partners (%) | 64.4 (2014) | [13] |
| Condom use: MSM, commercial (%) | 58.7 (2014) | [12] |
| Condom use: FSW, casual partners (%) | 78.1 (2014) | [12] |

(*Continued*)

**Table 1.** (Continued)

| Input | Value | Source |
|-------|-------|--------|
| Condom use: FSW, commercial (%) | 80.7 (2014) | [12] |
| Condom use: Clients, casual partners (%) | 61.9 (2014) | [12] |
| Condom use: Clients, commercial (%) | 51.6 (2014) | [12] |

lower than that for established pediatric (US$410.0). Personnel costs accounted for the largest proportion of the mean cost PPY for pre-ART (41%) and both pediatric ART types (39% each), followed by supplies and utilities. ARV accounted for 29% and 32% of the mean cost PPY for newly initiated adults and established adults, while personnel costs accounted for 29% and 30%, respectively. The mean costs PPY per virally suppressed patient were US$647.6 (range: US$432.9 to US$1,482.0; SD: US$376.1).

## Site-level unit costs

A site-level analysis revealed that all unit costs of HTS and ART at stand-alone ART and TB/HIV facilities were lower than other types of facilities. There is high variation of the costs within private and integrated facilities. Private facilities had higher testing yield than public facilities (3.1% vs 2.4%), and incurred lower economic costs than all government-owned facilities combined, with costs per client tested of US$11.8 vs US$15.5 and per positive patient identified of US$446.2 vs US$896.1. Cost per client tested was similar in rural and urban sites (US$15.0 and US$14.5, respectively), but cost per positive client identified was substantially higher in rural sites (US$1,445.0 vs US$471.7) driven by lower yield at rural sites (1.6% vs 3.1%).

Private hospitals had higher economic ART costs PPY than all government-owned hospitals combined (US$711.1 vs US$347.9). These differences were relatively consistent across

**Table 2.** HIV investments in each program area in Guyana in 2015.

| Programmatic area | ASC or other category | Government of Guyana | US Government | Global Fund | Total |
|-------------------|----------------------|---------------------:|--------------:|------------:|------:|
| General population prevention | Not disaggregated | - | - | 211,996 | 211,996 |
| FSW program | 01.08 | 12,457 | 191,397 | 19,222 | 223,076 |
| MSM program | 01.09 | 28,476 | 108,727 | 19,222 | 156,425 |
| HTS program | 01.03, 02.01.01 | 195,795 | 456,460 | 78,850 | 731,105 |
| ART program | ARV | - | 173,563 | 291,956 | 465,519 |
| | ART site-level service delivery | 923,072 | 395,602 | - | 1,318,675 |
| | Other USG spending on ART program | - | 385,313 | | 385,313 |
| | ART central operations (MoPH, NAPS, NPHL) | 187,384 | - | - | 187,384 |
| Lab | 02.01.05 | 736,297 | 238,616 | - | 974,913 |
| PMTCT | 01.17.01, 01.17.02, 01.17.98 | 295,419 | 110,337 | - | 405,756 |
| Other prevention | 01.19, 01.22.98, 01.07, 01.98 | 137,954 | 245,752 | - | 383,706 |
| Other care | 02.01.04, 02.01.07, 02.01.09, 02.01.98 | 142,681 | 508,414 | 118,858 | 769,953 |
| Management | 04.01, 04.02, 04.04 | 127,524 | 2,757,326 | - | 2,884,850 |
| Orphans and vulnerable children | 03.01, 03.02, 03.03 | - | 149,665 | - | 149,665 |
| Monitoring & evaluation | 04.03, 04.05, 04.06 | 32,190 | 216,779 | - | 248,969 |
| Infrastructure | 04.07, 04.08, 04.10.01 | 4,762 | 389,519 | - | 394,281 |
| Human resources | 05.02, 05.03, 05.98 | 256,890 | 125,928 | - | 382,818 |
| Enabling environment | 07.03, 07.98 | - | 184,087 | - | 184,087 |
| Research | 08.98 | - | 121,951 | - | 121,951 |
| **TOTAL** | | **3,080,901** | **6,759,436** | **740,104** | **10,580,442** |

**Table 3. Selected characteristics of 9 study sites.**

|  | Number of health facilities |
|---|---|
| Region |  |
|  2 | 1 |
|  3 | 1 |
|  4 | 5 |
|  6 | 1 |
|  10 | 1 |
| Type of administration |  |
|  Government | 7 |
|  Private | 2 |
| Location |  |
|  Urban | 7 |
|  Rural | 2 |
| Model of services |  |
|  Integrated (ART, PMTCT and other health services) | 6 |
|  TB/HIV | 1 |
|  Stand-alone ART clinic | 1 |
|  Stand-alone PMTCT | 1 |

various cost inputs (per-patient personnel costs, equipment costs, etc.). Private facilities had higher cost per virally suppressed patient (US$935.2 vs US$729.2) although viral suppression rates were higher in private than government facilities (78.0% vs 61.6%). Costs PPY on ART in rural facilities were higher than in urban settings (US$477.6 vs US$404.1), driven by personnel and laboratory costs. Costs per virally suppressed patient were similar in rural and urban facilities ($781.4 vs $790.7), with viral suppression rates slightly higher in rural facilities (71.0% vs 62.3%).

To account for variation in costs and client volume, we calculated the weighted average cost for HTS, pre-ART and ART (Table 6) by multiplying site-level unit costs by client volume, adding them up, and divided the costs by the total volume. We applied a 3% discount rate to estimate the net present value per patient, and lifetime costs of one HIV infection of US$9,088.9.

**Table 4. Costs per client tested (2016 USD).**

| Input type | Range | Mean |
|---|---|---|
| **Recurrent costs** |  |  |
|  Personnel | 2.8–22.5 | 9.1 |
| Supplies | 0.8–4.4 | 1.7 |
|  Building use | 0.1–2.1 | 0.6 |
|  Utilities | 0.5–8.3 | 1.8 |
|  Contracted services | 0.1–4.2 | 0.6 |
| **Capital costs** |  |  |
|  Equipment | 0.1–2.1 | 0.6 |
|  Training | 0.1–0.9 | 0.1 |
|  New infrastructure[1] | 0.7 | 0.1 |
| **Total** | 4.7–31.4 | 14.7 |

[1] Only one study site incurred new infrastructure costs.

**Table 5. Mean cost per patient year by input and patient type (2016 USD).**

| Input type | Pre-ART | Newly initiated adult ART | Established adult ART | Newly initiated pediatric ART | Established pediatric ART |
|---|---|---|---|---|---|
| **Recurrent costs** | | | | | |
| Personnel | 131.3 | 138.2 | 127.2 | 149.5 | 158.3 |
| ARV | | 136.2 | 136.2 | 21.2 | 21.2 |
| Other drugs | 5.6 | 5.6 | 5.6 | 5.6 | 5.6 |
| Lab supplies | 117.3 | 112.6 | 101.3 | 124.0 | 134.2 |
| Other supplies | 23.8 | 34.6 | 13.3 | 30.1 | 47.1 |
| Building use | 5.7 | 5.1 | 5.1 | 5.4 | 5.4 |
| Utilities | 15.8 | 16.4 | 16.4 | 15.3 | 15.3 |
| Contract services | 10.1 | 10.1 | 10.1 | 10.1 | 10.1 |
| **Capital costs** | | | | | |
| Equipment | 8.8 | 8.8 | 8.8 | 8.9 | 8.9 |
| Training | 3.7 | 3.7 | 3.7 | 3.7 | 3.7 |
| ARV buffer stocks | - | 0.4 | 0.4 | 0.0* | 0.0* |
| Total (SD) | 322.2 (181.8) | 471.8 (272.3) | 428.2 (171.7) | 374.0 (220.3) | 410.1 (312.2) |

*0.01.

## Resource requirements and optimal resource allocation

If 2015 investment patterns had continued, we estimate that Guyana would not have been likely to attain the targeted 50% reduction in HIV incidence by 2020 compared to 2012 levels [12]. There would be 660 new infections and 170 HIV-related deaths in 2020 corresponding to a 17% increase from 2012 estimated new infections (Fig 1A). However, if some investments had been shifted away from regions 7, 8, 9 and 10, and redistributed to the highest-burden regions (Regions 2 and 4; Fig 1B), then both HIV incidence and mortality would have declined

**Table 6. Cost estimates and related assumptions for modeling.**

| | Value | Source |
|---|---|---|
| *One-off costs* | | |
| Weighted average cost per client tested | US$10.8 | Cost estimates |
| Weighted average cost per positive patient identified | US$431.8 | Cost estimates |
| *Ongoing costs* | | |
| Weighted average cost per patient retained in pre-ART care | US$271.0 | Cost estimates |
| Weighted average cost per patient retained in ART | US$396.6 | Cost estimates |
| *Duration of ongoing costs* | | |
| Duration of pre-ART care | 0.5 years [0 years-1 years] | Assumption |
| Average cost per patient retained in ART | 35 years [20 years-50 years] | Assumption |
| *Net present value of ongoing costs (3% discounting)* | | |
| Net present value of pre-ART care | US$135.5 [US$0-US$271.0] | Calculation |
| Net present value of ART | US$8,521.6 [US$5,900.3-US$10,204.2] | Calculation |
| *Lifetime costs of one infection* | | |
| Total | US$9,088.9 [US$6,332.1-US$10,907.0] | Calculation |

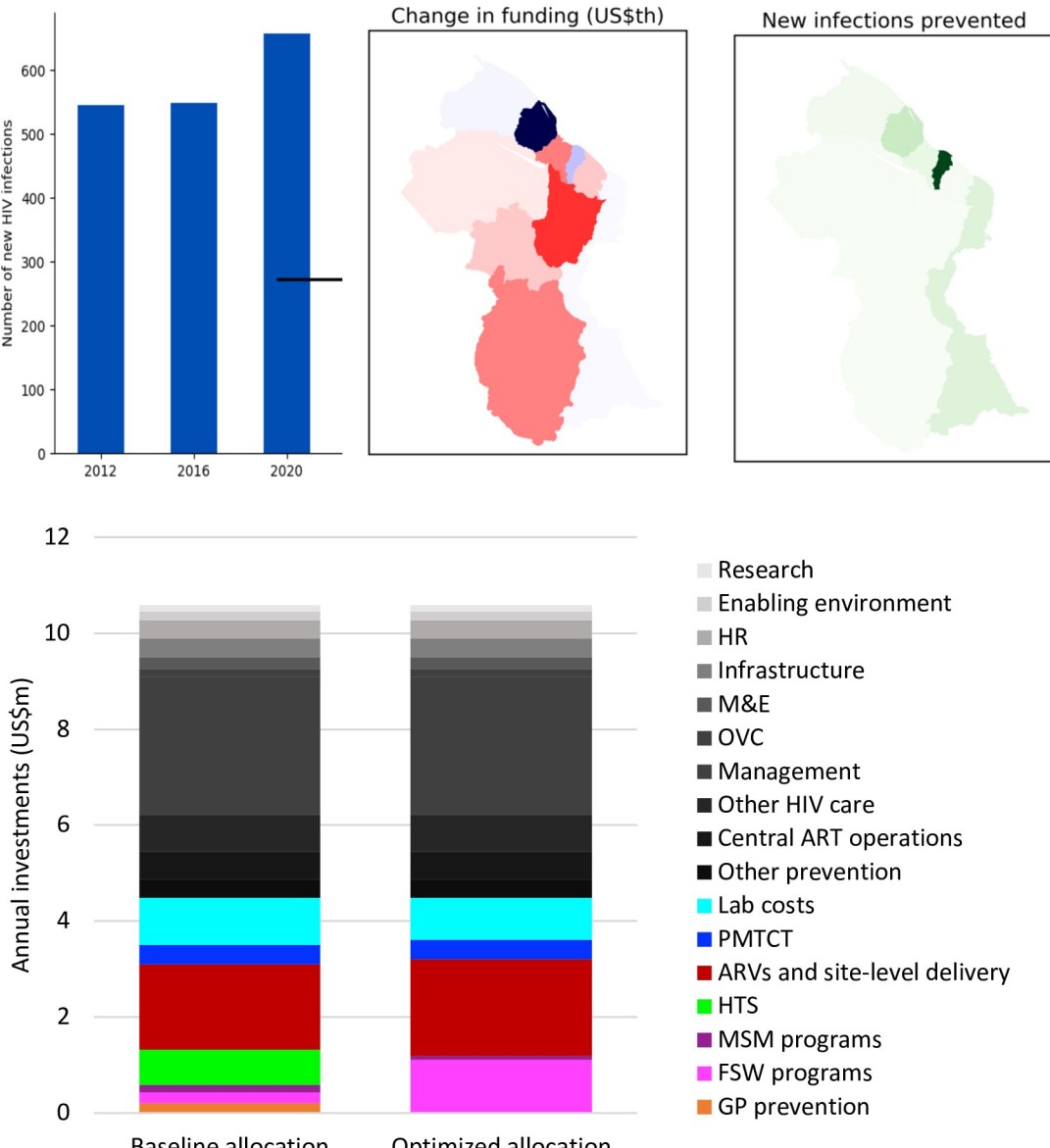

**Fig 1. Summary of the baseline epidemic outcomes, along with the funding changes recommended in order to get as close as possible to the NSP targets without increasing investments.** (A) New infections in Guyana in 2012, 2016 and 2020 (year by which NSP targets are to be reached), assuming that the 2015 budget is annually available and distributed as it was in 2015; (B) the model-recommended changes in investment per region; (C) the number of HIV infections that would be averted if funds were optimally allocated in Guyana; (D) a comparison of baseline and optimal allocations for each program.

further, with the majority of averted infections in Regions 2 and 4 (Fig 1C). By program, the highest priority would have been to scale up treatment and care programs: the model's optimization algorithm estimates that even if no additional resources were made available for the HIV response, increasing investments in the ART program by 13% from US$1.8m to US$2.0m annually (Fig 1D) would have been the most cost-effective way to reduce new infections and control the epidemic and to ensure that all PLHIV have access to quality care. We also estimate that epidemic gains could have been possible if spending on primary prevention programs were re-prioritized. Whilst the population of FSW is not estimated to be large, the

**Table 7. Summary of the results of the modeling exercise, comparing the investment levels and epidemiological and economic outcomes of core scenarios.**

| | Scenario | Average annual investments 2017–2020 | | | | | | Cumulative outcomes 2017–2020 | | |
|---|---|---|---|---|---|---|---|---|---|---|
| | | ARV & site-level ART delivery (a) | All other core HIV programs (b) | All core programs (c = a+b) | All non-targeted programs (d) | Total HIV budget (c +d) | Increase relative to 2015 HIV budget | HIV infections averted relative to scenario 1 (%) | HIV-related deaths averted relative to scenario 1 (%) | New lifetime care costs averted relative to scenario 1 (US$m) |
| 1) | Budget allocated as in 2015 (no additional resources) | US$1.8m | US$2.7m | US$4.5m | US$6.1m | US$10.6m | US$0 (0%) | - | - | - |
| 2) | Budget allocated optimally (no additional resources) | US$2.0m | US$2.5m | US$4.5m | US$6.1m | US$10.6m | US$0 (0%) | 37% | 22% | US$7.4m |
| 3) | Budget increased in order to achieve NSP targets, allocated as in 2015 | US$5.9m | US$8.9m | US$14.8m | US$6.1m (assumed not to increase) | US$20.9m | US$10.3m (98%) | 50% | 38% | US$9.9m |
| 4) | Budget increased in order to achieve NSP targets, allocated optimally | US$3.8m | US$5.1m | US$8.9m | US$6.1m (assumed not to increase) | US$15.0m | US$4.4m (42%) | 50% | 38% | US$9.9m |

results of the optimization analysis nevertheless indicate that investing in programs for FSW would be more cost-effective than investing in prevention programs for the general population. If funds were reallocated across programs and regions as depicted in Fig 1C and 1D, then compared to a baseline in which funds were allocated as in 2015, an additional 37% of new infections could have been averted between 2017 and 2020, which would in turn avert US $7.4m in lifetime treatment costs (Table 7, scenario 2).

We estimate that Guyana could have achieved its NSP 2020 target of a 50% reduction in new infections (compared to 2012 levels) by doubling investments and redistributing funds across regions and programs (Fig 2). This would have required annual investments of US $8.9m across 7 core programmatic categories (general population prevention programs, FSW programs, MSM programs, HTS, ART, prevention of mother to child transmission (PMTCT) and lab monitoring/retention). Achieving the NSP targets would have required Guyana to more than double investments in the ART (from annual investments of US$1.8m to US $3.8m), to scale up HTS by ~20% (from annual investments of US$0.7m to US$0.9m), and to scale up prevention programs for both FSW (by tenfold) and MSM (by fourfold). By contrast, it would have been more expensive to achieve the NSP targets if the 2015 funding patterns had continued, requiring annual investments of around US$14.8m in the above core programmatic categories (Fig 2). Meeting the NSP targets would have resulted in estimated savings of US$9.9m lifetime treatment costs (Table 7, scenarios 3 and 4).

## Discussion

This study is the first analysis of costs along the HIV cascade, HIV resource requirements and efficiency of HIV response in Guyana and the Caribbean. As expected, costs per client tested were driven by volume and costs per positive identified were driven by testing yield. The cost variation by site could be partly attributable to variation in volume. There is some evidence of economies of scale in HTS (cost per client tested tends to be lower in high volume facilities), but there is no such evidence in the ART services. The cost variation may also be due to facility characteristics and client needs. Nevertheless, there is scope for further investigations to assess

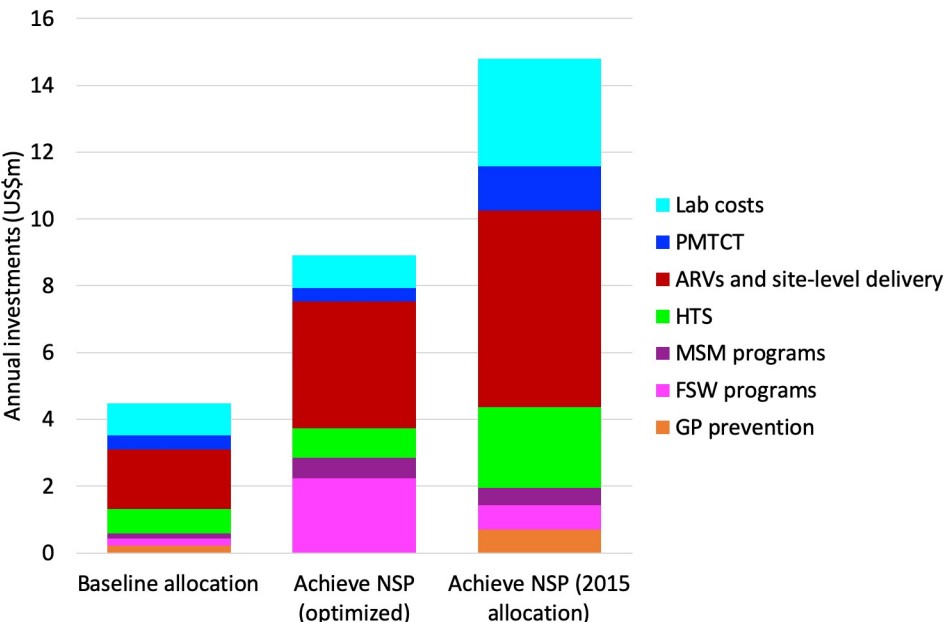

**Fig 2. The left bar shows the allocation of funds across 7 core programmatic categories in 2015.** The middle bar shows the "optimized" allocation, i.e., the allocation estimated to the minimum required for achieving the NSP targets. The right bar shows the funding levels that would be required to meet the NSP targets if the 2015 allocation of funds continued.

whether the sites with lower unit costs may be harnessing efficiencies that other sites could also exploit.

Given the relatively high costs associated with identifying new positive cases, concentrating efforts on successfully linking HIV-positives to ART, retaining patients on ART and improving viral suppression through adherence activities will be critical in lowering overall program costs and improving efficiency. Finding positives will become more challenging as the proportion of PLHIV aware of their status increases, leading to lower testing yields.

The unit costs of HIV testing services found in this study are lower than those found in other studies conducted in middle-income countries, but the costs per positive identified are markedly higher [22]. These costs are driven by personnel, a finding common to other studies [22,23]. Personnel costs also make up a large proportion of the unit costs of ART (30% and 40% for adult and pediatric patients respectively), but ARVs and laboratory costs are also important contributors, with laboratory costs in particular contributing 24% and 33% for adult and pediatric patients. These costs were based on the Guyana's HIV treatment guidelines, which recommend at least two CD4 tests and two VL tests annually. As guidelines are updated, lower costs associated with lab testing due to less number of tests conducted may be expected depending on the scale of lab operations.

We attempted to cost community-based services and linkage services to provide further insights on the HIV services, but community-based services had not yet formally implemented, and data on community-based services and linkages were unavailable. Implementing a reporting and information system to record and monitor the number of patients successfully linked to care and virally suppressed patients will enhance the ability to monitor key outcomes and potentially identify more efficient service delivery models. Considering implementation of differentiated service delivery models among stable and unstable patients may lower the costs and improve efficiency of service delivery.

This study has several limitations. The estimated costs PPY for virally suppressed patients may be less reliable because data on the number of virally suppressed patients were not recorded and maintained at all study sites. The estimates of the lifetime treatment costs averted do not consider the off-setting savings that may result from providing effective treatment, such as having less frequent episodes of illness or hospitalization and being better able to continue working or undertaking other productive activities. The modelling component of this study is subject to the usual limitations of epidemic modelling studies, including sensitivity to data inputs (which can be unreliable, biased and/or missing) and sensitivity to the modelling framework used (which is an imperfect representation of true epidemic dynamics). The key inputs to the epidemiological (as summarized in Table 1) were only available as point estimates, and since confidence intervals were not available, we therefore chose not to attempt to quantify the uncertainty around the model outputs (including estimates of HIV infections and deaths). The model's results should therefore be interpreted with caution, since no formal statistical tests could be conducted to compare epidemiological outcomes under different funding scenarios. The modeling results are also based on the assumption of ongoing funding availability, which may not eventuate if donor funding continues to decline and government funding does not replace it.

Our finding that there is scope for allocative efficiency improvements in Guyana is consistent with HIV allocative efficiency studies in other parts of the world [24–27]. Relatively little work has been done on the allocative efficiency of HIV responses in the Caribbean, although there have been studies of regional spending patterns [28], resource needs [29] and sustainability of HIV responses in the region, particularly in the context of donor withdrawal [30–32]. In agreement with these studies, we find that more resources would be needed to meet the targets associated with ending AIDS. However, there is cause for optimism as well, with scope for allocative efficiencies that would reduce the extent of resources required.

## Conclusions

Understanding and quantifying the resources needed for scaling up HIV services in targeted locations and populations, as well as identifying options for maximizing the impact of HIV investments, will help guide strategies for the national HIV response. In a context of limited resources, the high variation in service costs by facility coupled with the scope for allocative efficiency improvements call for better targeting of services and efficient service delivery models to improve yields and maximize program outcomes.

## Acknowledgments

We are grateful to the Guyana Ministry of Public Health and management teams and staff of health facilities for their contributions to the study. We thank Dr. Colin Roach, Pamela Joseph, Anya Krivelyova and Kristine Allen for their support, and Cliff Kerr for preliminary work on the modeling analyses. The findings and conclusions in this paper are those of the authors and do not necessarily represent the official position of the funding agencies.

## Author Contributions

**Conceptualization:** Chutima Suraratdecha, Rachel Albalak.

**Data curation:** Rachel Albalak.

**Formal analysis:** Robyn M. Stuart.

**Funding acquisition:** Chutima Suraratdecha, David P. Wilson.

**Methodology:** Robyn M. Stuart.

**Project administration:** Chutima Suraratdecha.

**Supervision:** David P. Wilson.

**Validation:** Morris Edwards, Rhonda Moore, Nadia Liu.

**Writing – original draft:** Chutima Suraratdecha, Robyn M. Stuart.

**Writing – review & editing:** Chutima Suraratdecha, Robyn M. Stuart, Morris Edwards, Rhonda Moore, Nadia Liu, David P. Wilson, Rachel Albalak.

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
