## [Decision Letter · Decision Letter 0]

19 May 2020

PONE-D-19-34434

Costs of providing HIV care and optimal allocation of HIV resources in Guyana

PLOS ONE

Dear Dr Stuart, 

Thank you for submitting your manuscript to PLOS ONE. After careful consideration, we feel that it has merit but does not fully meet PLOS ONE’s publication criteria as it currently stands. Therefore, we invite you to submit a revised version of the manuscript that addresses the points raised during the review process.

We would appreciate receiving your revised manuscript by Jul 03 2020 11:59PM. To enhance the reproducibility of your results, we recommend that if applicable you deposit your laboratory protocols in protocols.io, where a protocol can be assigned its own identifier (DOI) such that it can be cited independently in the future. For instructions see: http://journals.plos.org/plosone/s/submission-guidelines#loc-laboratory-protocols

We apologize for the delay in the review process, this has been delayed due to the Covid19 Pandemic. We look forward to receiving your revised manuscript.

Kind regards,

Eduard J Beck, PhD, FAFPHM, FFPH, FRCP

Academic Editor

PLOS ONE

Journal Requirements:

2. Please remove the 'Draft' watermark from all the way through your manuscript.

Reviewers' comments:

Reviewer's Responses to Questions

**Comments to the Author**

1. Is the manuscript technically sound, and do the data support the conclusions?

Reviewer #1: Yes

Reviewer #2: Partly

2. Has the statistical analysis been performed appropriately and rigorously? 

Reviewer #1: I Don't Know

Reviewer #2: Yes

3. Have the authors made all data underlying the findings in their manuscript fully available?

Reviewer #1: Yes

Reviewer #2: Yes

4. Is the manuscript presented in an intelligible fashion and written in standard English?

Reviewer #1: Yes

Reviewer #2: Yes

5. Review Comments to the Author

Reviewer #1: While variance in cost between sites may be an opportunity for efficiency, I wondered if different clinics served different populations, had different viral load suppression rates and as such had different human resource needs.

Consideration could be given to including the limitations of this analysis . Eg are we assuming that loss to follow up in this analysis is included or excluded.

Reviewer #2: This is an interesting paper that reports in great detail the results of a microcosting exercise to calculate unit costs along the HIV care cascade in Guyana. The unit costs are then used to parametrise a resource allocation model looking at two possible scenarios (historical budget allocation vs reallocation to optimise incidence and mortality outcomes). While the paper’s reporting on the costing exercise methods, results and possible limitations is thorough and comprehensive, there is comparatively quite limited space devoted to explaining the methods and assumptions underlying the resource allocation modelling.

There are two main concerns in this respect. Firstly, it is not clear why the authors chose to explore those two particular scenarios in the model. Optimal vs historical allocation seem to be the best and worst case, respectively, which is definitely interesting for exploring any resource gaps. However, there is no indication as to whether any of these scenarios is actually realistic or perhaps other more realistic scenarios could have been explored. Did the authors consider expert elicitation or validate these scenarios with local stakeholders in the budgeting process? And, if not, how were the scenarios chosen and why? Perhaps the overall aim of the analysis should be clarified, particularly in terms of the perspective and intended audience for the allocative efficiency analysis. In the discussion, the it is stated that the analysis shows that there are vulnerabilities in the HIV financing model in Guyana due to reliance on donor funding (lines 263-264). This is not emerging clearly in the analysis and perhaps could have been further explored in the model?

Secondly, there is no mention throughout the manuscript of how uncertainty was treated in the model and no intervals are reported for the epidemiological parameters and results. Although the analysis uses a published, referenced model, this information is vital for assessing the reliability of model results.

Other suggestions for improvement:

Line 41: the term PLHIV was never defined

Lines 64-65: were the facilities randomly selected?

Line 91: capital costs is a more standard term than “investment costs” and it is used below. I suggest changing to capital costs in every instance for clarity and consistency

Line 123: “geographical optimization algorithm”, fine to refer to source publication but perhaps helpful for readers to include here a brief explanation, at least of the criteria used to optimise

Line 170: “weighted average”, consider mentioning the weighting formula in the methods

Lines 218-221: this sentence is quite long and may be missing some punctuation

In general, there are several unclear sentences that appear to be missing punctuation or clauses

Table 5 is not easy to read and, in general, presenting median costs is not a standard requirement. I would suggest removing the median and presenting mean and standard deviation on the same line, as well as switching rows and columns in the table to make it more compact

6. PLOS authors have the option to publish the peer review history of their article (what does this mean?). If published, this will include your full peer review and any attached files.

Reviewer #1: No

Reviewer #2: No

---

## [Author Response · Author response to Decision Letter 0]

4 Jun 2020

We have uploaded a response letter file separately.

---

## [Decision Letter · Decision Letter 1]

19 Aug 2020

Costs of providing HIV care and optimal allocation of HIV resources in Guyana

PONE-D-19-34434R1

Dear Dr. Stuart,

We’re pleased to inform you that your manuscript has been judged scientifically suitable for publication and will be formally accepted for publication once it meets all outstanding technical requirements.

Kind regards,

Kevin Lu, PhD

Academic Editor

PLOS ONE

Reviewers' comments:

Reviewer's Responses to Questions

**Comments to the Author**

1. If the authors have adequately addressed your comments raised in a previous round of review and you feel that this manuscript is now acceptable for publication, you may indicate that here to bypass the “Comments to the Author” section, enter your conflict of interest statement in the “Confidential to Editor” section, and submit your "Accept" recommendation.

Reviewer #2: All comments have been addressed

2. Is the manuscript technically sound, and do the data support the conclusions?

Reviewer #2: Yes

3. Has the statistical analysis been performed appropriately and rigorously? 

Reviewer #2: Yes

4. Have the authors made all data underlying the findings in their manuscript fully available?

Reviewer #2: Yes

5. Is the manuscript presented in an intelligible fashion and written in standard English?

Reviewer #2: Yes

6. Review Comments to the Author

Reviewer #2: Many thanks to the authors for addressing initial comments. The manuscript is now suitable for publication

7. PLOS authors have the option to publish the peer review history of their article (what does this mean?). If published, this will include your full peer review and any attached files.

Reviewer #2: No

---

## [Editor Report · Acceptance letter]

19 Oct 2020

PONE-D-19-34434R1 

Costs of providing HIV care and optimal allocation of HIV resources in Guyana 

Dear Dr. Stuart:

I'm pleased to inform you that your manuscript has been deemed suitable for publication in PLOS ONE. Congratulations! Your manuscript is now with our production department. 

Kind regards, 

on behalf of

Professor Kevin Lu 

Academic Editor

PLOS ONE